# Influence of Interactions between Nitrogen, Phosphorus Supply and *Epichloё*
*bromicola* on Growth of Wild Barley (*Hordeum brevisubulatum*)

**DOI:** 10.3390/jof7080615

**Published:** 2021-07-29

**Authors:** Mingxiao Lang, Jingle Zhou, Taixiang Chen, Zhenjiang Chen, Kamran Malik, Chunjie Li

**Affiliations:** State Key Laboratory of Grassland Agro-Ecosystems, Key Laboratory of Grassland Livestock Industry Innovation, Ministry of Agriculture and Rural Affairs, Engineering Research Center of Grassland Industry, Ministry of Education, Gansu Tech Innovation Centre of Western China Grassland Industry, Center for Grassland Microbiome, College of Pastoral Agriculture Science and Technology, Lanzhou University, Lanzhou 730000, China; langmx19@lzu.edu.cn (M.L.); jinyy16@lzu.edu.cn (J.Z.); chentx@lzu.edu.cn (T.C.); chenzhj17@lzu.edu.cn (Z.C.); malik@lzu.edu.cn (K.M.)

**Keywords:** *Hordeum brevisubulatum*, *Epichloë* endophyte, nitrogen, phosphorus supplements, nutrient stoichiometry, plants growth

## Abstract

*Epichloë* endophytes are biotrophic fungi that establish mutualistic symbiotic relationship with grasses and affect performance of the host under different environments. Wild barley (*Hordeum brevisubulatum*) is an important forage grass and often infected by *Epichloë* *bromicola*, thus showing tolerances to stresses. Since the plant growth correlates with both microbial infection and nutrient stoichiometry, this study was performed to investigate whether the function of *Epichloë* *bromicola* endophyte to improve host growth depend upon the nitrogen (N), phosphorus (P) fertilization. *Epichloë*-infected (E+) and *Epichloë*-free (E−) wild barley plants were subjected to nine types of mixed N (0.2 mM, 3 mM, 15 mM) and P (0.01 mM, 0.1 mM, 1.5 mM) levels treatments for 90 d to collect plant samples and determine multiple related indexes. We found that *E. bromicola* and N, P additions positively affected seed germination. Further, *E. bromicola* significantly enhanced chlorophyll content and root metabolic activity under N-deficiency, and meanwhile, might alter allocation of photosynthate under different conditions. The contents of N, P and stoichiometry of C:N:P of E+ plants were significantly higher than that of E− under nutrient deficiency, but contrary results were observed under adequate nutrients. Therefore, we propose that the growth-promoting ability of *E. bromicola* is closely correlated with N and P additional levels. Under low N, P additions, positive roles of endophyte are significant as opposed to negative roles under high N, P additions.

## 1. Introduction

Nitrogen (N) and phosphorus (P) are two crucial nutrient elements considered to influence the plant survival. As the major elements, N is the key ingredient of pivotal molecules like amino acids, amides, and nucleic acids, whereas P is an essential component of ATP, phosphates, and ribosomes [1]. Previous studies suggested that nutrient stoichiometry and allocation are related to the plant strategies including the balance of multiple key elements, growth, vegetation composition, and nutrient condition at the community level [2]. As the principal elements of plants, the carbon (C), N, P, and their ratios (C:N:P) have the mechanistic link with the growth rate of organisms [3]. Plants growth, dynamic balance of soil fertility, alteration of competitive relationship, and capabilities of growth and development can be reflected by the ratios of C:N, C:P, and N:P [4,5,6]. The dynamics of N and P thus affect both structures and functions of plants. In addition, N and P play essential roles in fungal survival [7], and besides providing the requirements of growth as other organisms, they are important in regulating metabolism. For example, nitrogen catabolite produced by endophyte depends on a nitrogen source and nutrient utilization. N and P were both involved in alkaloid synthesis induced by endophyte, which protect the host plants from insects and intake of grazing livestock [8]. Thus, understanding how plants perform in response to N, P, and endophyte can bring new sight to create cultivation of plant varieties.

The establishment of symbiotic relationships in plants with microorganism is a pivotal advancement of evolution [9]. *Epichloë* endophytes (formerly genus *Neotyphodium*) are endosymbiotic fungi, and colonize in host plants without showing apparent symptoms [10]. These endophytes often infect temperate grasses of the Pooideae subfamily by systematically colonizing their mycelium in the intercellular spaces of host plants and spread in all nutrient-rich tissues except roots [11,12]. Host species have symbiotically conferred resistance against, including drought, heavy metals, salinity, cold, allelopathy, pathogens and insects [13,14,15]. *Epichloë* endophytes provide certain benefits to host grass in seed germination, plant height and photosynthesis [16]. Likewise, some *Epichloë* endophytes endow the host with enhanced tolerance to poor nutrients by increasing the nutrient absorption and utilization efficiency of the host [17]. However, the positive mutualistic relationship is usually implicated in the plant growth phase, environment, and genotypes of host and endophyte [18]. Change of symbiotic relationship, such as switch from mutualism to parasitism, may be due to environmental conditions [19].

Plant root has various functions including acquiring water and nutrients, perceiving the change of soil environment and assimilating, transporting amino acids and plant hormones, etc. [20]. Nutrient uptake traits play an essential role in plants to avoid imbalance of nutrients by regulating nutrient concentration in tissues [21]. Previous studies showed that the presence of *Epichloë* endophyte affect root morphology and biomass, inducing a range of root metabolic activities that could influence nutrient uptake, although mycelium of *Epichloë* endophyte has never been observed in roots [18,22]. Therefore, C, N, and P, as crucial nutrients in soil, are usually affected by *Epichloë* endophyte inevitably. However, there is no consistent regulation for a certain element, which is mainly linked to the species of host, endophyte, and their environment [8,23]. For instance, *Epichloë* endophyte infection can significantly increase N and P contents in wild barley under salt stress [4], and promote the growth and increase soluble carbohydrate content of perennial ryegrass, but this benefit switches under the poor N environment [18]. Applications of exogenous N and P such as inorganic fertilizers are main measures for host grasses in terms of high yield as well as good quality [22,24]. However, the elevated usage of chemical fertilizer alone aggravates burdens to the ecosystem, such as soil acidification, water eutrophication, environmental pollution and even damage to human health [22]. Meanwhile, unbalanced input of inorganic fertilizer has a serious influence on ecological stoichiometry and even the functions of ecosystems [22,25]. The balance of nutrient elements rather than the content of a single element for plants to uptake may contribute more to sustainable development for aquatic ecosystems [26] and land plant communities [27]. For example, N availability may alter utilization efficiency of other nutrients such as P and potassium (K) [28], while the limitation of N can switch to the deficiency of P after the application of nitrogen fertilizer [29]. Experimental manipulation of ratios and concentrations of exogenous N and P can help to improve the understanding of mechanism caused of *Epichloë* endophytes in plant nutrient acquisition in response to varying environment [30]. However, there is no direct report related to the strength of exogenous combination of N and P supplements with *Epichloë* endophytes on their host plants. From the results of our previous research, we hypothesized that: (1) there are differences between different concentrations of N and P application and *Epichloë bromicola* infection in terms of host plants germination and growth. (2) The benefits of *E. bromicola* on host plants depend on exogenous N and P conditions.

Wild barley (*Hordeum brevisubulatum*) is a common forage grass with high yield and superior quality that is widely adaptable to a range of stress environments, including barrenness, salt, alkali, cold, and drought [25]. *H. brevisubulatum* grows in Linze County, Gansu province, China, and is often infected with the *E. bromicola* at an infection rate of 80–90%, approximately [31]. In our study, nine groups of N and P combinatorial nutrient solutions were prepared to simulate different nutrient supplements. *H. brevisubulatum* with (E+) or without (E−) *E. bromicola* were treated in these solution, and investigated the effect of exogenous N and P supplements on symbiotically conferred strength, by analyzing growth parameters of wild barley.

## 2. Materials and Methods

### 2.1. Plant Material

Seeds of *H. brevisubulatum* with mature stems were collected from the Linze Experimental Station of Lanzhou University (100°06′55″ E, 39°11′89″ N, altitude 1350 m), Gansu Province, China, in July 2017. The region has an annual average temperature and precipitation of 7.6 °C and 117 mm, respectively. The endophyte infection status (E+ or E−) was determined by the microscopic examination of aniline blue stained culms (Carl Zeiss Suzhou Co., Ltd., Suzhou, China) based on 10× and 40× [32]. All acquired seeds were stored at 4 °C to maintain endophyte viability.

### 2.2. Experimental Design

#### 2.2.1. Preparation of Nutrient Solution

Nutrient solution was configured to nine types with three different concentrations of N (0.2 mM, 3 mM, 15 mM) and P (0.01 mM, 0.1 mM, 1.5 mM), respectively: high nitrogen and high phosphorus (HNHP), high nitrogen and medium phosphorus (HNMP), high nitrogen and low phosphorus (HNLP), medium nitrogen and high phosphorus (MNHP), medium nitrogen and medium phosphorus (MNMP), medium nitrogen and low phosphorus (MNLP), low nitrogen and high phosphorus (LNHP), low nitrogen and medium phosphorus (LNMP), and low nitrogen and low phosphorus(LNLP), which are presented in Table 1. The solution was 1/2 strength modified Hoagland nutrient solution, ammonium nitrate (NH_4_NO_3_) and potassium dihydrogen phosphate (KH_2_PO_4_) were used as sources of N and P, respectively, Potassium chloride (KCl) was added to maintain the equal K^+^ content in different solution.

#### 2.2.2. Seed Germination Experiment

Three thousand seeds were germinated on filter paper in 54 Petri dishes (9 cm in diameter) (27 for E+ and 27 for E−) with 50 seeds per dish, after surface sterilization. Seeds were immersed for 3 min in 75% alcohol, then the seeds were rinsed 3 times with demineralized sterile water. The Petri dishes contain E+ and E− were divided into 9 groups and each type corresponding to N/P treatment with 3 replicates. A 4 mL corresponding N/P nutrient solution was supplied. The germination conditions in artificial climate box were set to 25 °C/15 °C (day/night), 10-h light/14-h dark photoperiod, a light intensity of 50 μmol m^−2^ s^−1^, supplying equal quantity of water in Petri dish to maintain the moist environment. To calculate the germination rate (%), germination potential (%), and germination index, the number of seeds with a ruptured seed coats (pointing radicle emergence) were recorded daily. Radicle and embryo length was measured at the end of experiment (15 days). All the seed parameters were calculated according to the International Seed Testing Association (ISTA) protocols (ISTA 1999).

#### 2.2.3. Hydroponic Experiment

A hydroponic experiment was performed in a greenhouse of Yuzhong Experimental Station of Lanzhou University, Gansu Province. Seeds obtained from E+ and E− plants were surface sterilized and sown in a germination tray filled with sterilized vermiculite. After germination for two weeks, uniform seedlings were transplanted into plastic boxes (17.5/12.5 upper/lower diameter × 12 cm high; 6 seedlings per box), and each box was filled with 1200 mL different types of nutrient solution and subjected to N/P supplements. The boxes that contained E+ and E− seedlings were divided into 9 groups, respectively, with 6 replicates. The growth conditions in the greenhouse were as follows: 27 °C/23 °C for day/night, 12-h/12-h for light/dark cycle, with light supplied at 120 μmol m^−2^ s^−1^. The nutrient solution in each container was refreshed after every 7 days to ensure nutrient concentration. All plants were harvested after 90 days.

#### 2.2.4. Plant Biomass and Physiological Indexes

Shoot height and tiller number were measured before reaped by direct measurement and counting method, respectively. All plants were separated into shoots and roots, rinsed with distilled water after harvested. To determine the dry weight, all samples were oven-dried at 80 °C, until reach to a constant weight. Chlorophyll content was determined by direct extraction method (0.2 g leaf samples were leached in 80% acetone overnight until the tissue turned colorless, the homogenate was centrifuged to get the upper solution, absorption spectra of the solution measured by spectrophotometer (Model-721, Shanghai Precision and Scientific Instrument Co., Shanghai, China) at 645 (A_645_), 663 (A_663_) and 652 nm (A_652_), with the 80% acetone as blank control) [33]. Root metabolic activity was measured using the method of triphenyl tetrazolium chloride (TTC) [34]. In brief, 0.5 g root sample was placed in 10 mL beaker, 10 mL 0.4% TTC solution and 10 mL phosphoric acid buffer (pH = 7.0). Sample was immersed completely in the mixture solution and under dark condition at 37 °C for 3 h, afterwards the reaction was stopped by adding 2 mL 1 mol L^−1^ sulfuric acid (H_2_SO_4_). The water of the root, which was collected from the mixed solution, was absorbed, and grounded with 4 mL ethyl acetate and a little quartz sand in mortar to gain triphenyl formazan (TF). The absorbance of TF and blank control were determined at 485 nm, and the root TTC reductive number was calculated with standard curve. The formula used was: TTC intensities (mg g^−1^ h^−1^) = TTC reductive amounts (mg)/[root weight (g) × time (h)].

#### 2.2.5. Chemical Analysis

Plant organic carbon (OC) content was measured using K_2_CrO_7_-H_2_SO_4_ oxidation method (oil bath at 175 ± 5 °C for 5 min, followed by titration with FeSO_4_) [35]. For total nitrogen (N) and total phosphorus (P) contents, dried samples were digested using H_2_SO_4_ with CuSO_4_ and K_2_SO_4_ (CuSO_4_:K_2_SO_4_ = 1:10 mixture) catalyzing under a digestion block at 420 °C for 1 h. Then, the N and P concentrations were determined by a flow injection analyzer (FIAstar 5000 Analyzer, Foss, Denmark). The proportion of C, N and P (C:N, C:P and N:P) were calculated.

Dried samples of plant tissues were dissolved with a tri-acid (HNO_3_:H_2_SO_4_:HClO_4_ = 8:1:1) treatment to estimate the mineral content. The concentration of Na^+^ and K^+^ were measured with flame spectrometer (M410, Sherwood, Britain).

### 2.3. Statistical Analysis

Data analyses were performed with SPSS version 20.0 (SPSS, Inc., Chicago, IL, USA). Three-way AVOVA was used to determine the effects of N supply (N), P supply (P) and *E. bromicola* (E) on germination parameters, the length of root and embryo, biomass, C, N, P contents, Na^+^, K^+^ concentrations, and ratios of C:N, C:P, N:P. Significance of difference tests between E+ and E− plants under the same N and P supply conditions in all of the parameters were determined using independent *t*-test. The parameters of seeds and plants were log-transformed to meet the normality assumption of ANOVA. Statistical significance was defined at the 95% confidence level. Data in figures were showed as means with their standard errors (SE).

## 3. Results

### 3.1. Seed Germination

The interaction of N, P, and endophyte showed no significant differences in seed germination (*p* > 0.05) and embryo growth (*p* = 0.051). Radicle length was influenced by N, P, *E. bromicola*, and their interaction (Appendix A). E+ seeds suffered less than the E− seeds under LN (0.2 mM) during germination (Figure 1). Both N and *E. bromicol**a* endophyte promoted germination and the interaction of N, P, and endophyte caused a more significant impact on radicle length (*p* < 0.001) than embryo length (*p* = 0.051) (Appendix A). We conclude that the interaction of N and P supplements with endophyte significantly promotes germination under poor nutrient condition and the early development of roots.

### 3.2. Growth and Physiological Parameters

Underground biomass was significantly affected by the interaction of N and P supplements with endophyte (*p* < 0.001). Irrespective of *E. bromicola* presence, N, P, and their interaction had positive impacts on plant growth, as showed by the increment in the growth parameters recorded for the biomass, plant height, and tiller number (Appendix A). The underground biomass of E− plant was significantly higher than E+ under LNLP (0.2 mM, 0.01 mM), LNMP (0.2 mM, 0.1 mM) and HNHP (15 mM, 1.5 mM), but in E+ plants, it was higher than E− plants only under HNMP (15 mM, 0.1 mM) (Figure 2a). The infection of endophyte significantly affected the plant height and was higher in E+ plants than E− plants under HNLP (15 mM, 0.01 mM), MNMP (3 mM, 0.1 mM) and MNHP (3 mM, 1.5 mM) (Figure 2b).

Results showed that the interaction of N, P, and endophyte significantly influenced the root metabolic activity (*p* = 0.010). Meanwhile, this interaction also significantly increased the chlorophyll content and enhanced root metabolic activity (Appendix A). Under LN (0.2 mM) and HN (15 mM) treatments, chlorophyll content of E− plants was positively correlated with the concentration of P, and that of E+ plants increased with a rise in N concentration under three P levels (Figure 3a). Chlorophyll content of E+ plants was significantly higher than E− plants under LP (0.2 mM) and LNMP (0.2 mM, 0.1 mM), and the same results were observed in HP (1.5 mM) treatment, although in this case the differences were not significant. The root metabolic activity of the E− plants was increased as the P concentration grew under MN (3 mM) and HN (15 mM). A significantly higher root metabolic activity was recorded in E+ plants than E− plants under MNLP (3 mM, 0.01 mM) and HNLP (15 mM, 0.01 mM) (Figure 3b).

### 3.3. C, N, and P Contents

The contents of C, N, and P in leaves were significantly affected by N and P supplements, endophyte, and their interaction (*p* < 0.05) (Table 2). Significant decrease of C content in E+ leaves was observed with increasing P concentration under HN (15 mM), and rose in both leaves and roots with N concentration under MP (0.1 mM) and LP (0.01 mM). Endophyte infection strengthened the increase in content of C in leaves and roots under LP (0.01 mM). C content of E+ plants in leaves was higher than E− under HNLP (15 mM, 0.01 mM) and HNMP (15 mM, 0.1 mM), lower than E− under LN (0.2 mM) (Figure 4a). As for N content in leaves, it was significantly lower in E− than in E+ under LN (0.2 mM), and higher in E− than in E+ under HN (15 mM) (Figure 4b). Comparison of E+ and E− plants demonstrated that endophyte infection alleviated the increased trend of P content in roots. Under HP (1.5 mM) treatments, the E− leaves had higher P content vs. the E+, while contrary results was observed when exposed to LP (0.01 mM) treatments (Figure 4c).

Contents of N and P in roots were significantly influenced by N and P supplies, endophyte, and their interaction (*p* < 0.05) (Table 2). N content of E+ plants in roots decreased with increasing P concentration under LN (0.2 mM) and MN (3 mM), additionally, in E− plants showed an increased trend as increased N concentration gradually under MP (0.1 mM) and HP (1.5 mM). The N content in roots was higher in E+ than in E− plants under LN (0.2 mM), with contrary results was observed under HN (15 mM) (Figure 4a). For P content in roots, which in E− increased with increasing P concentration under three levels of N, and in E+ declined as increasing N concentration under LP (0.01 mM), MP (0.1 mM), and HP (1.5 mM). In addition, the P content showed sharp increased pattern with endophyte infection under LNLP (0.2 mM, 0.01 mM), LNMP (0.2 mM, 0.1 mM), MNLP (3 mM, 0.01 mM), and MNMP (3 mM, 0.1 mM), that in E+ was significantly lower than E− under MNHP (3 mM, 1.5 mM) HNHP (15 mM, 1.5 mM), and HNMP (15 mM, 0.1 mM) (Figure 4c).

### 3.4. Stoichiometric Ratios of C, N, P

C:N, C:P, and N:P were significantly affected by N, P, endophyte, and their interaction, except for the effect of endophyte on C:P in roots (*p* = 0.762) (Table 3). As observed for C:N ratio of E− in leaves and roots increased as P concentration increased under LN (0.2 mM), meanwhile, the ratio was significantly higher than E+. Additionally, there was decreased patterns of C:N ratio on E+ in leaves and roots with increasing P concentration, and that of E+ was higher than E− under HN (15 mM) (Figure 5a). Declined trend was observed in C:P ratio of E− plants as P concentration increased under LN (0.2 mM), MN (3 mM) and HN (15 mM), while increased trend was found on E+ plants in C:P ratio with increasing N concentration under LP (0.01 mM) and MP (0.1 mM) (Figure 5a). Comparison of E+ and E−, C:P ratio was significantly higher in E− than in E+ under LNLP (0.2 mM, 0.01 mM), LNMP (0.1 mM, 0.1 mM), MNLP (3 mM, 0.01 mM), and MNMP (3 mM, 0.1 mM). Under HP (1.5 mM), and C:P ratio of E+ plants was higher than E− plants (Figure 5b). For the ratio of N:P, E− plants in leaves were positively affected by N supply and negatively influenced by P supply. N:P of E+ in plants was observed with a decreased trend as increasing N supply under HP (1.5 mM) as compared to an increased trend of E−. In comparison of E+ and E−, N:P of E+ in both leaves and roots was higher than that of E− under LN (0.2 mM) and HP (1.5 mM) (Figure 5c).

### 3.5. Na^+^, K^+^ Contents

Na^+^ contents in leaves and roots and K^+^ contents in roots were significantly affected by the interaction of N and P supply with endophyte (*p* < 0.05). Additionally, the difference of Na^+^ and K^+^ caused by N, P, endophyte, and their interaction were more apparent in roots than in leaves (Appendix A). Irrespective of endophyte infection, Na^+^ and K^+^ in roots showed increased trends with increasing N and P supplements under MP (0.1 mM), HP (1.5 mM), MN (3 mM), and HN (15 mM). In general, E+ had higher Na^+^ content under LP (0.01 mM), and lower Na^+^ content than that of E− under HN (15 mM) and HP (1.5 mM), although the difference was not significant in any case (Appendix A). Leaves of E− had higher K^+^ than E+ under HNLP (15 mM, 0.01 mM). Under LNMP (0.2 mM, 0.1 mM), MNMP (3 mM, 0.1 mM) and MNHP (3 mM, 1.5 mM), E+ in leaves had more K^+^ content. Further, addition of P had a positive effect on K^+^. The strength of *E. bromicola* on K^+^ was apparent under moderate nutrient additions. Generally, E− in roots had higher K^+^ content than E+ (Appendix A).

## 4. Discussion

This study is one of the few focused on the interaction between N and P supplied with *E. bromicola* on their host plants-*H. brevisubulatum*. The results indicate that the combination of a suitable quantity of N and P additions with *E. bromicola* was beneficial to seed germination and plants growth. The positive effectiveness of *E. bromicola* on plants was more apparent in nutrient deficiency condition by improving chlorophyll content, root metabolic activity, and promoting absorption of N and P to maintain a lower C:N and C:P. Under sufficient nutritional conditions, the morphological index of wild barley sharply increased. *E. bromicola* can regulate the Na^+^ content of roots to maintain it at a relatively stable level. A high nutrient environment reduces the beneficial role of *E. bromicola* on wild barley.

### 4.1. Effect of Epichloë Endophyte Infection on Seed Germination and Growth of Wild Barley

Endophyte infection could contribute to agronomic advantages and affect metabolic processes of host plants [15]. The impact of *Epichloë* infection to promote seed germination under abiotic stresses has been determined in many studies [16]. Similar results were found in the present study that endophyte ameliorated seed germination under low fertilizer conditions. It had been demonstrated that the expression of many genes associated with protein turnover were regulated in seeds by endophyte, which were implicated in accelerating protein translation and degradation. Stressed condition could increase the level of protein synthesis; therefore, defensive proteins were produced to enhance tolerance of plants to adverse environments [36]. It was a possible coping strategy adopt by endophytes to combat nutrient deficiency in seeds of wild barley. Several mechanisms have been reported to explain the finding that *Epichloë* endophyte can improve growth of host plants. On the one hand, a study related to root metabolic activity, mentioned the intensity of root growth, nutrient exudation for rhizobacterial symbiosis and nutrient uptake [37]. On the other hand, N content in leaves affect the photosynthesis [38]. Variation of photosynthetic pigments concentration is similar to photosynthetic capability [39]. Our data showed that *E. bromicola* positively affected chlorophyll content and root metabolic activity. We thus infer that enhanced chlorophyll content is associated with the increased root metabolic activity, driven by endophyte. However, current results showed that no significant differences were observed in total biomass, plant height and tiller numbers between E+ and E− plants. It is possible that sensitivity to endophyte infection on physiological index is stronger than morphological index. To better understand positively effects of *Epichloë* endophyte, the underlying mechanisms between metabolic products, protein and endophyte during early seed germination and seedling growth is much needed to explore.

Chen et al. found that endophyte affected contents of C, N, and P, which involved in metabolic process of synthesis [40]. Previous studies reported that endophyte could enhance the activities of nitrogen metabolism enzymes [41], intrinsic competitive ability and nutrient acquisition [42]. In addition, alkaloids might be related to availability of N and P [43]. Results from the present study support these finding that endophyte ameliorated N and P contents in both leaves and roots under N and P deficiency. Thus, it can be concluded that endophytes enhance fertilizer utilization under poor nutrient conditions, which is in accordance with the results of Lewis et al. [44]. Alternatively stating, alkaloids were synthesized in E+ as protective secondary metabolites under environmental stresses. However, endophytes caused no significant difference on C content in plants. This may be because, C has several pathways for assimilation by plants which were different from N and P [45]. Thus, it is possible that N and P contents in E+ and E− plants are main elements to influence the growth and metabolism caused by *E. bromicola* in *H. brevisubulatum.* Thereby affecting patterns of resource allocation as well as metabolic activity are possible coping strategies for *E. bromicola* to influence *H. brevisubulatum* growth.

### 4.2. Effect of N and P Supplements on Seed Germination and Growth of Wild Barley

Nutrients are crucial for plants growth, developments and crop yield. A previous study discussed that nitrogen compounds stimulate pentose phosphate pathway or induce formation of enzyme that inactivate abscisic acid (ABA), and are accompanied by related metabolic pathway in seeds [46]. The addition of N positively affected the seed germination and soaking the seeds into nitrogen compounds before sowing, is a common measure to improve germinating power [47].

Dual nutrients supplements of N and P are important to obtain a broad picture of difference of mechanisms on plants response to main nutrients [15]. N and P function in plant metabolic activities by affecting gene expression, photosynthesis, sugar metabolism, carbohydrate allocation among plant organs and reproductive activities [22,48]. Resource allocation in plants between aboveground and belowground is flexible in response to various environments, which was reflected by biomass. Previous reports showed that plants increase the allocation of photosynthetic products to roots to obtain N under N-deficiency conditions [46]. When nutrients are not limited, plants allocate more energy to the aboveground portion to capture maximum light energy [49]. In this study, exogenous N and P addition significantly increased the biomass of both leaves and roots, plant height, and tillers numbers. Meanwhile, the root:shoot ratio was higher under high nutrient treatments than limited conditions. Specially, these indexes had sharp increased trends with increasing P or N addition under abundant nutrient conditions as opposite to the gentle variation when there was N or P-deficiency, which partly explained the allocation mechanism above [50]. Extensive study suggested that nutrients are important for root growth and metabolic activity. Our study showed that P addition promoted root metabolic activity of E−, especially when N was not limited. Additionally, the strength of one nutrient depends on the availability of another [50]. An alternative explanation is that P or N uptake transport systems are generally up-regulated by N or P addition [47]. Our results showed that endogenous N and P contents in both leaves and roots of E− were positively correlated with N and P supplements, similar to the response of non-legumes plants to improved N and P supplements artificially [51]. P or N contents in plants significantly enhanced as another nutrient addition escalate under abundant nutrient supplements. These synergistic growth responses may attribute to organic P storage, P recycling, and P uptake [52]. In brief, allocation of nutrients to cellular components and physiological function in plants were affected by corresponding variation of N and P contents [53].

### 4.3. Effect of Interaction of N, P Supplements and Epichloë Endophyte Infection on Seed Germination and Growth of Wild Barley

Plants can regulate growth patterns and resource requirements in response to various environments. For example, increased root hair density was observed under P-deficiency [52] and root physiological metabolism increased when availability of P and K improved [22]. Some studies suggested that endophyte infection contributed to ameliorate adaptation of hosts to environments [4,40]; however, the degree of growth plasticity on fungal symbionts regulation is still unclear. 

C, N, and P are the foundation of biochemical compositions for life. Previous studies had demonstrated that the presence of *Epichloë* endophyte changed C, N, and P contents of host plants under barren soil [22]. Our results confirmed our hypothesis that endophyte alters the response of wild barley to N and P addition, and the direction caused by endophyte partly depends on nutrient conditions. Further, the allocation of photosynthates to roots was affected by endophyte to obtain more nutrients from surroundings under limited nutrients conditions. More photosynthates were accumulated in leaves when nutrients were abundant. Interestingly, higher N and P concentrations were available in aboveground and belowground structure in E+, compared to E− under LN (0.3 mM) and LP (0.01 mM), respectively. Nevertheless, contrary results were observed under HN (15 mM) and HP (3 mM), respectively, which was consistent with the result of Lewis et al. [42], one alternative or additional explanation is endophyte improved P uptake under limited P, and increased N fertilizer use efficiency to response N-deficiency stress [8,44]. This has significant implication for the host since this flexible allocation mechanism may contribute to plants growth in response to variation of nutrient supply. Further, excessive nutrients addition may be allocated to synthesis of alkaloid and defensive compounds rather than to growth [54].

There is an intrinsic linkage between plant growth rate and tissue elemental stoichiometry [4]. Hessen et al. proposed growth rate hypothesis (GRH) that growth rate had a negative correlation with ratios of C:N, C:P, and N:P. Organisms accommodate changes in growth by regulating their C:N:P, in order to adapt to varying environments [6]. Plants with low C:N and C:P ratios have strong capacity to absorb nutrients and accumulate C [55]. Plants with an excellent growth advantages in poor nutrients conditions are those that can regulate their own nutrient contents and increase nutrient efficiency without lowering their growth rates [54]. In the present study, E+ plants had lower C:N and C:P ratios than E− in both leaves and roots under N and P-deficiency, and the results were contrary with abundant nutrient supplements. However, the beneficial function of endophyte in limited nutrient conditions was to transform into inhibition in abundant nutrient supplements. Variation of P in plants is caused by utilization of P to ribosomal RNA (rRNA). This mechanism is required to provide protein synthesis needs for increased growth rate [6]. It partly explained the linkage of plant growth and variation of P content and C:P in our research. The value of N:P is regarded as an indicator to judge the adaptability of plant growth to nutrient condition [22]. Our results showed that endophyte infection alleviated N and P restriction of plants growth under limited nutrient conditions, which was necessary for photosynthesis and protein synthesis to meet the strong growth rate of plants [6]. This shown the changes of growth rate driven by endophyte infection on *H. brevisubulatum* under different nutrient conditions. Taken together, we conclude that the magnitude and direction of endophyte strength effects on nutrient stoichiometry varied with N and P additions.

Na^+^ and K^+^ are involved in energy metabolism, material transport, etc. Effects of *Epichloë* endophyte on Na^+^ and K^+^ content focus on salt stress. Further, *Epichloë* endophyte had significant influence on Na^+^ and K^+^; however, this strength appeared only under salt-stress [43]. The results of the present study indicated that the function of interaction of N and P and endophyte on regulating Na^+^ and K^+^ in roots was stronger than leaves. These phenomena could be explained that roots directly interact with soil nutrients. Although the changes driven by *E. bromicola* infection on Na^+^ and K^+^ contents in different nutrient conditions were irregular. In addition, Na^+^ content maintained stable relatively regulated by endophyte with increasing fertilizer in roots. This might be because Na^+^ and K^+^ are involved in the regulatory mechanism in homeostasis management of endophyte in response to the external nutrition environment. We infer that there are similar regulated mechanisms of Na^+^ and K^+^ between nutrient conditions and salt-stresses, and the variation may be linked with growth stages.

From these findings, we conclude that the response of *H. brevisubulatum* with *E. bromicola* infection to different exogenous N and P addition may be multifaceted, including physiological processes, nutrient stoichiometry, photosynthesis, and mineral elements. Endophyte has the strength on host plants to response to various environments. We can better predict that how fertility of soil, even species abundances and dominant species, change via analyzing complex variation of interaction between endophyte and plants in the future [56].

## 5. Conclusions

In this study, we investigated the possible roles of interaction between *Epichloë* endophytes infection and N and P supplements on wild barley germination, plants growth, chlorophyll content, root metabolic activity, nutrient contents and stoichiometry. The presence of *Epichloë* endophyte improved germination characteristics and radicle growth. The possible mechanisms by which *Epichloë* endophyte infection enhanced the growth of plants under lower fertilizer include increasing chlorophyll content, root metabolic activity and regulating nutrient contents and ratios, as well as mineral elements. Our finding also confirmed that the effect of *Epichloë* endophyte depends on the levels of N and P supplements, as excessive supplements of N or P decreased the positive function of endophyte. Collectively, based on the knowledge of the interaction between N and P fertilizer and endophyte, we provide a theoretical and practical basis for wild barely infected *Epichloë* endophyte to improve their performance with a rational ratio of N and P fertilizers.

## Figures and Tables

**Figure 1 jof-07-00615-f001:**
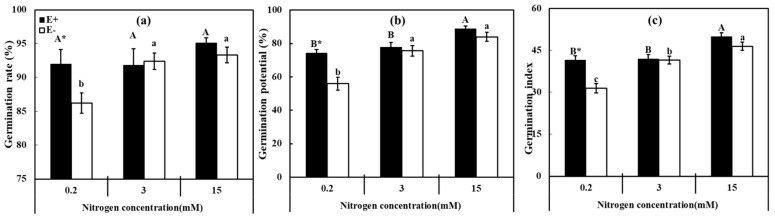
Germination parameters of *Hordeum brevisubulatum* under different concentrations of N. (**a**) Germination rate, (**b**) germination potential, and (**c**) germination index. E+: *Epichloë bromicola*-infected plants, E−: *E. bromicola*-free plants. Different lowercase letters and majuscule letters on top of bars indicate significant differences (*p* < 0.05) of E− and E+ plants between the different N concentrations, respectively. An * on the top of the bars, means a significant difference at *p* < 0.05 (independent *t*-test) between E+ and E− plants under the same N concentration.

**Figure 2 jof-07-00615-f002:**
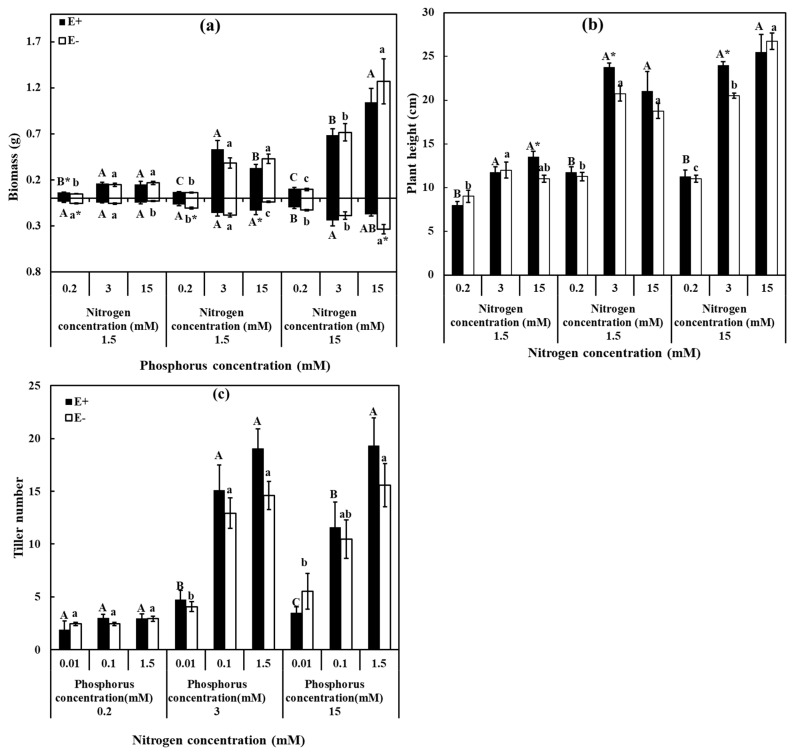
Growth parameters of *H. brevisubulatum* under different concentrations of N and P. (**a**) Biomass, (**b**) plant height, and (**c**) tiller number. E+: *E. bromicola*-infected plants, E−: *E. bromicola*-free plants. Different lowercase letters and majuscule letters on top of bars indicate significant differences (*p* < 0.05) of E− and E+ plants between the different N concentrations, respectively. An * on the top of bars, means significant difference at *p* < 0.05 (independent *t*-test) between E+ and E− plants under the same nutrient concentration.

**Figure 3 jof-07-00615-f003:**
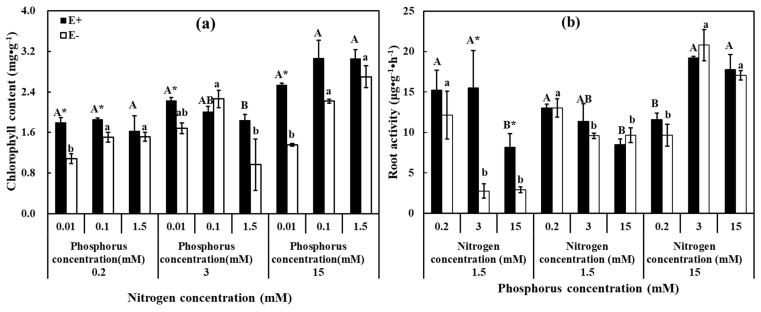
Chorphyll content and root metabolic activity of *H. brevisubulatum* under different concentrations of N and P. (**a**) Chorphyll content, (**b**) root metabolic activity. E+: *E. bromicola*-infected plants, E−: *E. bromicola*-free plants. Different lowercase letters and majuscule letters on top of bars indicate significant differences (*p* < 0.05) of E− and E+ plants between the different N concentrations, respectively. An * on the top of bars, means significant difference at *p* < 0.05 (independent *t*-test) between E+ and E− plants under the same nutrient concentration.

**Figure 4 jof-07-00615-f004:**
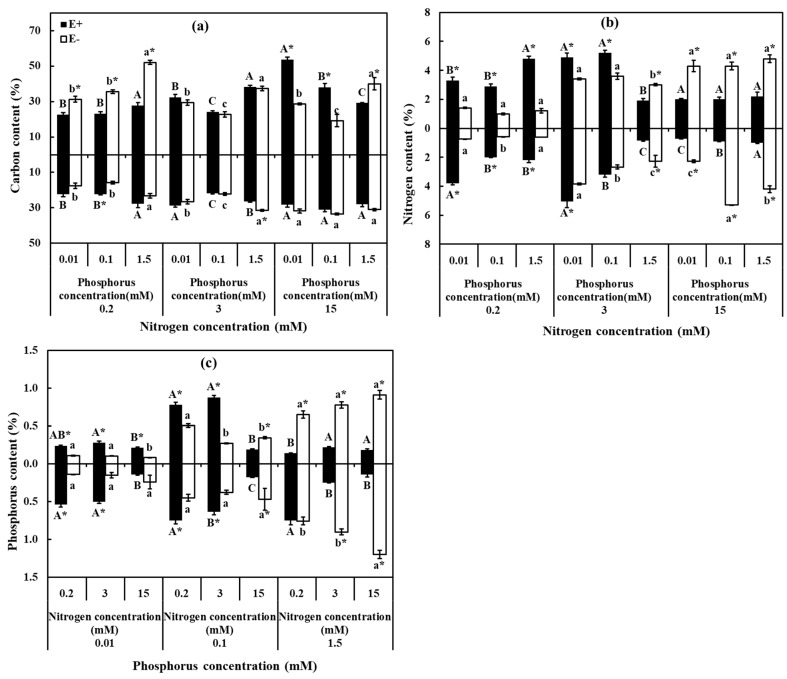
Carbon, nitrogen, and phosphorus contents of *H. brevisubulatum* under different concentrations of N and P. (**a**) Carbon content, (**b**) nitrogen content, and (**c**) phosphorus content. E+: *E. bromicola*-infected plants, E−: *E. bromicola*-free plants. Different lowercase letters and majuscule letters on top of bars indicate significant differences (*p* < 0.05) of E− and E+ plants between the different N concentrations, respectively. An * on the top of bars, means significant difference at *p* < 0.05 (independent *t*-test) between E+ and E− plants under the same nutrient concentration.

**Figure 5 jof-07-00615-f005:**
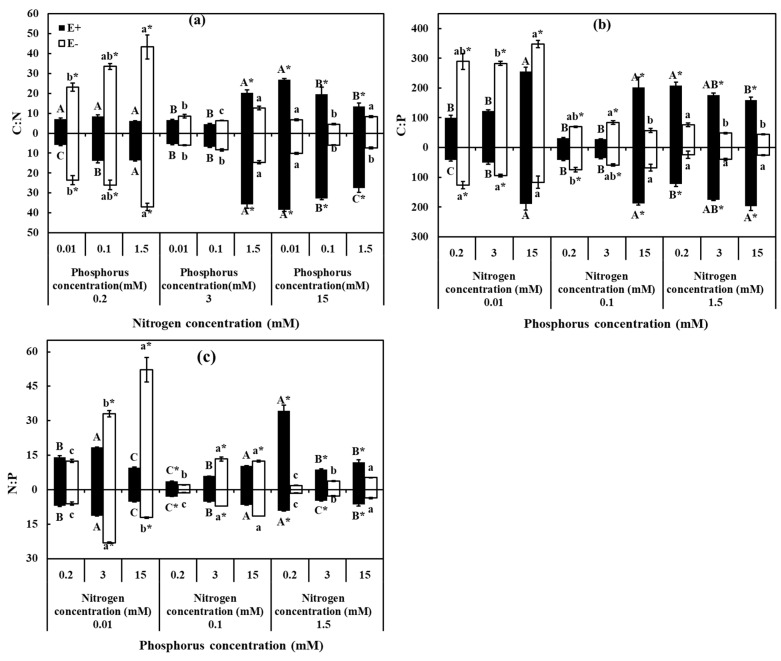
The stoichiometric ratios C, N, and P of *H. brevisubulatum* under different concentrations of N and P. (**a**) C:N, (**b**) C:P, and (**c**) N:P. E+: *E. bromicola*-infected plants, E−: *E. bromicola*-free plants. Different lowercase letters and majuscule letters on top of bars indicate significant differences (*p* < 0.05) of E− and E+ plants between the different N concentrations, respectively. An * on the top of bars, means significant difference at *p* < 0.05 (independent *t*-test) between E+ and E− plants under the same nutrient concentration.

**Table 1 jof-07-00615-t001:** The N and P concentrations of nutrient solution used for both experiments (mM).

	HNHP	HNMP	HNLP	MNHP	MNMP	MNLP	LNHP	LNMP	LNLP
Nitrogen	15.00	15.00	15.00	3.00	3.00	3.00	0.20	0.20	0.20
Phosphorus	1.50	0.10	0.01	1.50	0.10	0.01	1.50	0.10	0.01

**Table 2 jof-07-00615-t002:** Three-way ANOVA for the effects of endophyte (E), nitrogen concentration (N), and phosphorus concentration (P) on C, N, and P contents in leaves and roots of *Hordeum brevisubulatum*. N × P: interaction of N and P, N × E: interaction of N and *Epichloë bromicola**;* P × E: interaction of P and *E.*
*bromicola**;* N × P × E: interaction of N, P and *E.*
*bromicola*.

	Treatments	dF	Carbon Content		Nitrogen Content		Phosphorus Content	
			F	*p*	F	*p*	F	*p*
Aboveground	N	2	10.245	<0.001	130.149	<0.001	24.503	<0.001
	P	2	63.507	<0.001	59.87	<0.001	117.828	<0.001
	E	1	1.04	0.315	22.591	<0.001	31.596	<0.001
	N × P	4	25.314	<0.001	127.55	<0.001	10.9	<0.001
	N × E	2	108.468	<0.001	494.878	<0.001	111.535	<0.001
	P × E	2	57.469	<0.001	86.206	<0.001	139.707	<0.001
	N × P × E	4	18.014	<0.001	17.51	<0.001	9.807	<0.001
Underground	N	2	102.024	<0.001	62.323	<0.001	20.652	<0.001
	P	2	15.147	<0.001	2.528	0.093	255.71	<0.001
	E	1	0.647	0.426	6.856	0.013	30.452	<0.001
	N × P	4	14.937	<0.001	35.062	<0.001	48.885	<0.001
	N × E	2	22.853	<0.001	225.816	<0.001	50.025	<0.001
	P × E	2	2.404	0.105	1.718	0.193	385.567	<0.001
	N × P × E	4	1.845	0.142	18.922	<0.001	24.548	<0.001

**Table 3 jof-07-00615-t003:** Three-way ANOVA for the effects of nitrogen concentration (N), phosphorus concentration (P), and endophyte (E) on C:N, C:P, and N:P in leaves and roots of *H. brevisubulatum*. N × P: interaction of N and P, N × E: interaction of N and *E.*
*bromicola**;* P × E: interaction of P and *E.*
*bromicola**;* N × P × E: interaction of N, P and *E.*
*bromicola*.

	Treatments	dF	C:N		C:P		N:P	
			F	*p*	F	*p*	F	*p*
Aboveground	N	2	49.046	<0.001	41.894	<0.001	25.333	<0.001
	P	2	10.785	<0.001	316.173	<0.001	216.773	<0.001
	E	1	18.402	<0.001	0.125	0.726	11.111	0.002
	N × P	4	14.618	<0.001	27.108	<0.001	60.294	<0.001
	N × E	2	170.645	<0.001	29.508	<0.001	133.427	<0.001
	P × E	2	8.672	<0.001	236.95	<0.001	219.077	<0.001
	N × P × E	4	10.371	<0.001	13.654	<0.001	39.311	<0.001
Underground	N	2	95.05	<0.001	69.46	<0.001	551.407	<0.001
	P	2	35.921	<0.001	23.858	<0.001	764.766	<0.001
	E	1	49.63	<0.001	43.483	<0.001	42.373	<0.001
	N × P	4	61.98	<0.001	6.562	<0.001	389.111	<0.001
	N × E	2	484.522	<0.001	40.37	<0.001	131.882	<0.001
	P × E	2	4.697	0.016	69.82	<0.001	168.172	<0.001
	N × P × E	4	19.702	<0.001	2.754	0.044	119.217	<0.001

## Data Availability

Data are contained within the article.

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
