# Peer review of "Influence of Interactions between Nitrogen, Phosphorus Supply and Epichloёbromicola on Growth of Wild Barley (Hordeum brevisubulatum)"

_jof, 2021, doi:10.3390/jof7080615_

Round 1

Reviewer 1 Report

Keywords are repeated throughout the title of the work, please correct. The introduction lacks information on the influence of N and P on the development of fungi.

Line 95 to 100 - an excerpt from the methodology.

On line 104, seconds are missing from the GPS data. What microscope was used to make the endfit and what magnification Why couldn't I go to Epichloe - but would it be classic? Figures are not clearly, they should be corrected.

Remove or add spaces between values and units (depending on the requirements of the editorial office). Why has the effect of potassium (K) not been investigated as well?

Author Response

Dear the reviewer 1 and Editor:
Thanks for your approval and suggestions concerning our manuscript. Those comments are precious in improving our manuscript. We have studied comments carefully and made revisions accordingly which we hope it meets with your approval. Please see the attachment.

Reviewer 2 Report

Manuscript jof-1316105-peer-review-v1

“Interactions of Nitrogen, Phosphorus Supply and EpichloŃ‘ endophyte on Growth of Wild Barley (Hordeum brevisubulatum)”, J. Fungi.

This article demonstrates advances in the study of the effect of Epichloë sp. endophyte and nitrogen and / or phosphorus supplementation on various physiological parameters of wild barley (Hordeum brevisubulatum). The authors demonstrated that in some cases Epichloë sp with the addition of nitrogen or phosphorus is able to stimulate the germination and growth of wild barley, as well as influence some metabolic processes.

This paper demonstrates a good knowledge of current research on this topic. The authors provided an excellent introduction and discussion of the results obtained. However, before this manuscript can be published, some improvements should be performed:

General remarks:

  • There is problem with the title of the manuscript. I think authors should replace "interactions" with "influence".
  • The genus Epichloe is a group of phytopathogenic mycomycetes that parasitize plants of the cereal family. Determination of this endophyte to specific name could improve the work, otherwise it should be referred to in the manuscript as Epichloe sp.. I advise you to sequencing and determine the names of this endophyte down to the specific name.

Have other wild barley endophytes been found on your plants? At the same time, the plant can contain a wide variety of endophytes, which are also capable of influencing the development of the plant. It should be proved that it is Epichloe sp that affects the plant and not another endophyte. In this case, it is possible to use metagenomic sequencing. This can greatly enhance this work and influence the future of citations.

  • The manuscript contains a large number of tables and figures. I believe that some of the tables and figures can be represented in Supplementary Material or combined. In the description of the figures, it should be added that the data are presented as mean ± standard error (SE). The names of the tables should also be improved. In each table, it is necessary to decipher the abbreviations used in the table (this also applies to the figures).
  • It is also worth noting that this Ms contains errors in English grammar and writing style. I recommend that authors seek the help of a native English speaker or submit Ms to an English-speaking editorial service that reviews scientific articles.

Minor:

  • The vertical scale of Figures 2 (A), 4, 5, 6 shows values below zero, but I did not see a negative sign (-).
  • In Figures 2 (a, b) and 3 (b), two times different meanings have the same designations. If this is a mistake then it should be corrected.
  • Maybe you should shorten the discussion? This will improve the reader's perception.

Line 19: in abstract, add the designation (E+): Epichloë-infected plants, (E-): Epichloë-freeplants.

Line 126: add sterilization method or refer to other work.

Line 155: Replace «samples was» with «samples were».

Line 158: Which instrument is used to measure the absorption spectra? Indicate the name, manufacturer, city and country of the manufacturer.

Line 162: I propose to replace «stop the reaction» with «reaction was stopped».

Line 179: Indicate the manufacturer, city and of the manufacturer.

Line 226: I propose to replace « increasing» with «rised».

Line 230: I propose to replace « increasing» with «growed».

Line 253: I propose to replace « decreased C» with «decrease of C».

Line 256: I propose to replace « increase C content » with « increase in content of C».

Line 261: I propose to replace « of roots» with « in roots ».

Line 263: I misunderstood the sentence "Comparison between ...". I think this proposal is not correct. Please rephrase it.

Manuscript jof-1316105-peer-review-v1

“Interactions of Nitrogen, Phosphorus Supply and EpichloŃ‘ endophyte on Growth of Wild Barley (Hordeum brevisubulatum)”, J. Fungi.

This article demonstrates advances in the study of the effect of Epichloë sp. endophyte and nitrogen and / or phosphorus supplementation on various physiological parameters of wild barley (Hordeum brevisubulatum). The authors demonstrated that in some cases Epichloë sp with the addition of nitrogen or phosphorus is able to stimulate the germination and growth of wild barley, as well as influence some metabolic processes.

This paper demonstrates a good knowledge of current research on this topic. The authors provided an excellent introduction and discussion of the results obtained. However, before this manuscript can be published, some improvements should be performed:

General remarks:

  • There is problem with the title of the manuscript. I think authors should replace "interactions" with "influence".
  • The genus Epichloe is a group of phytopathogenic mycomycetes that parasitize plants of the cereal family. Determination of this endophyte to specific name could improve the work, otherwise it should be referred to in the manuscript as Epichloe sp.. I advise you to sequencing and determine the names of this endophyte down to the specific name.

Have other wild barley endophytes been found on your plants? At the same time, the plant can contain a wide variety of endophytes, which are also capable of influencing the development of the plant. It should be proved that it is Epichloe sp that affects the plant and not another endophyte. In this case, it is possible to use metagenomic sequencing. This can greatly enhance this work and influence the future of citations.

  • The manuscript contains a large number of tables and figures. I believe that some of the tables and figures can be represented in Supplementary Material or combined. In the description of the figures, it should be added that the data are presented as mean ± standard error (SE). The names of the tables should also be improved. In each table, it is necessary to decipher the abbreviations used in the table (this also applies to the figures).
  • It is also worth noting that this Ms contains errors in English grammar and writing style. I recommend that authors seek the help of a native English speaker or submit Ms to an English-speaking editorial service that reviews scientific articles.

Minor:

  • The vertical scale of Figures 2 (A), 4, 5, 6 shows values below zero, but I did not see a negative sign (-).
  • In Figures 2 (a, b) and 3 (b), two times different meanings have the same designations. If this is a mistake then it should be corrected.
  • Maybe you should shorten the discussion? This will improve the reader's perception.

Line 19: in abstract, add the designation (E+): Epichloë-infected plants, (E-): Epichloë-freeplants.

Line 126: add sterilization method or refer to other work.

Line 155: Replace «samples was» with «samples were».

Line 158: Which instrument is used to measure the absorption spectra? Indicate the name, manufacturer, city and country of the manufacturer.

Line 162: I propose to replace «stop the reaction» with «reaction was stopped».

Line 179: Indicate the manufacturer, city and of the manufacturer.

Line 226: I propose to replace « increasing» with «rised».

Line 230: I propose to replace « increasing» with «growed».

Line 253: I propose to replace « decreased C» with «decrease of C».

Line 256: I propose to replace « increase C content » with « increase in content of C».

Line 261: I propose to replace « of roots» with « in roots ».

Line 263: I misunderstood the sentence "Comparison between ...". I think this proposal is not correct. Please rephrase it.

Author Response

Dear the reviewer 2 and Editor:
Thanks for your approval and suggestions concerning our manuscript. Those comments are precious in improving our manuscript. We have studied comments carefully and made revisions accordingly which we hope it meets with your approval. Please see the attachment.

Round 2

Reviewer 1 Report

I accept the current version of the manuscript.